# Simulating the Effect of Gut Microbiome on Cancer Cell Growth Using a Microfluidic Device

**DOI:** 10.3390/s23031265

**Published:** 2023-01-22

**Authors:** Ekansh Mittal, Grace Cupp, Youngbok (Abraham) Kang

**Affiliations:** Department of Mechanical, Civil, and Biomedical Engineering, George Fox University, Newberg, OR 97132, USA

**Keywords:** gut microbiome, inflammation, microfluidics, cancer

## Abstract

The imbalance in the gut microbiome plays a vital role in the progression of many diseases, including cancer, due to increased inflammation in the body. Since gut microbiome-induced inflammation can serve as a novel therapeutic strategy, there is an increasing need to identify novel approaches to investigate the effect of inflammation instigated by gut microbiome on cancer cells. However, there are limited biomimetic co-culture systems that allow testing of the causal relationship of the microbiome on cancer cells. Here we developed a microfluidic chip that can simulate the interaction of the gut microbiome and cancer cells to investigate the effects of bacteria and inflammatory stress on cancer cells in vitro. To test the microfluidic chip, we used colorectal cancer cells, as an increased microbiome abundance has been associated with poor outcomes in colorectal cancer. We cultured colorectal cancer cells with Bacillus bacteria or lipopolysaccharide (LPS), a purified bacterial membrane that induces a significant inflammatory response, in the microfluidic device. Our results showed that both LPS and Bacillus significantly accelerated the growth of colorectal cancer cells, therefore supporting that the increased presence of certain bacteria promotes cancer cell growth. The microfluidic device included in this study may have significant implications in identifying new treatments for various cancer types in the future.

## 1. Introduction

The microbiome consists of trillions of beneficial and pernicious microorganisms colonizing the human body, and about 70% of those reside in the gut. These bacteria play a vital role in maintaining homeostasis, intestinal function, metabolism, absorption of drugs and nutrients, and many diseases in the body [1]. Emerging studies suggest that the gut microbiome plays a critical role in cancer initiation and progression [2,3,4]. Most of these studies show that an imbalance in the bacterial composition might be associated with increased cancer progression [5,6], possibly due to an increase in inflammation [7,8]. However, only limited studies have established a causal relationship [9,10,11]. Thus, it is important to clearly understand the relationship between the gut microbiome and cancer progression and investigate its mechanism in gut pathobiology. 

Many previous studies have used conventional culture methods, including multi-well culture plates, to co-culture bacteria and cancer cells, but these methods do not replicate the in vivo physiology and function that normally occur in living intestine [12,13,14]. It might also bring discrepancies between in vitro and in vivo testing. Microfluidic technologies have been adopted to develop in vitro physiological models by creating a tissue environment and its structural function [15,16,17]. Recently, many gut chips have been developed over two decades and gradually evolved from a simple 2D culture system to a complex 3D intestine tumor model. 

Over one decade ago, several groups developed the 2D gut chip that contains two independent channels sandwiched by a microporous membrane for culturing a single type of cells in a monolayer to evaluate the intestinal absorption rate of a drug [18,19]. Later, Ramadan et al. demonstrated the in vitro 2D intestinal barrier model by co-culturing the human colorectal adenocarcinoma cells with leukemia cells in a dual-layer [20]. Shah et al. created the 2D gastrointestinal human-microbe interface by co-culturing epithelial and microbial cells in a microfluidic chip [21]. Meanwhile, the 2D gut chips have been advanced for 3D culture and extracellular matrix (ECM) remodeling. Shim et al. demonstrated a 3D gut chip with a collagen scaffold to mimic the human intestinal villi in a microfluidic device [22,23]. Although they provided the 3D tissue structure and fluidic shear stress on the cell layer, they did not replicate mechanical motion. Recently, a gut chip with intestinal villi has been developed to create the intestinal lumen-capillary tissue interface in a microfluidic device [24,25,26]. The chip recapitulated cyclic mechanical motion and physiological fluid flow as occurring in vivo [27,28]. Moreover, Shin et al. co-cultured the Caco-2 cells with immune cells and gut microbiome to identify the initiator of the inflammatory host-microbiome cross-talk [29]. 

Most recently, there have been some attempts to show that cancer progression is associated with a microbiome while using a colorectal tumor chip [30,31] or during the testing drug efficacy [32,33]. Additionally, Gregorio et al. attempted to create an anaerobic cultural condition by generating an oxygen gradient along the thickness of the small intestine of the human tumor [34]. Maurer et al. presented the 3D tumor intestine on-chip to demonstrate the characterization of the immune response, microbial pathogenicity mechanisms, and quantification of cellular dysfunction attributed to alternations in the microbial composition [35]. Although these devices emulate tissue microenvironment and function, they have some limitations in simulating intestine pathophysiology, such as tumor hypoxic environment, or they are too complex for routine lab testing. Thus, there is still a need to develop not only a more physiologically relevant intestine tumor model to address specific research goals but also a novel strategy to investigate the relationship between the gut microbiome and cancer cells. 

Here, we demonstrated the effect of the gut microbiome on cancer cell growth using the in vitro tumor intestine chip to provide the physiological tumor hypoxic environment to co-culture human colorectal cancer cells with microbial cells. Our in vitro intestine model would be beneficial in investigating the interaction between pathogenic bacteria and drug or host and intestinal microorganisms and identifying the mechanisms of drug resistance for cancer and microbial cells. 

## 2. Materials and Methods

### 2.1. Fabrication of the Microfluidic Device

We designed a microfluidic device that consists of one middle chamber (width × length × height: 8000 × 20,000 × 250 µm) for cancer cell culture and two side channels (width × length × height: 1000 × 15,000 × 250 µm) to introduce media and bacteria (Figure 1A–C) [36]. The middle cell culture chamber has a rectangular shape with rounded corners so that fluid can be smoothly dispensed into the chamber and flow out of the outlet without any dead spots. The two side chambers are connected to the middle chamber by six linear channels with a width of 500 µm, a length of 1500 µm, and an angle of 60 degrees so that the gradient of substances, such as oxygen or bacteria, may be generated from one side to the other side in the middle chamber and the zonation of the gradient may maintain by continuous flow as minimizing shear stress on epithelial cells.

The template of the microfluidic device was made of SU-8 photoresist with 250 μm thickness on a silicon wafer using photolithography technology. The polydimethylsiloxane (PDMS) devices were molded from the SU-8 template according to the soft-lithography process [37]. Next, the microfluidic devices were bonded to a glass slide after air plasma treatment. The volume of the cell culture chamber in a microfluidic chip was approximately 160 µL. 

### 2.2. Co-Culture of Cancer Cells and Bacterial Cells

A human colorectal carcinoma cell line (red fluorescent protein (RFP)+ HCT116) was obtained from the Oregon Health and Science University. Colorectal cancer is highly abundant in Gram-positive (Streptococcus, Gemella) microflora with few Gram-negative (Fusobacterium) genus representations [38]. Therefore, to mimic the conditions of the gut microbiome, we selected one example of a Gram-positive (Bacillus) and Gram-negative (LPS, a purified bacterial membrane) stimulus. Testing with Bacillus requires a BSL-1 safety level, thus making testing conditions feasible. Further, Bacillus cells’ doubling time is approximately 2 h, and HCT116 cells’ doubling time is approximately 18 h. Thus, after initial optimization (data not shown) and to avoid the oversaturation of bacterial cells in microfluidics, we kept the ratio of bacteria and cancer cells at 1:40.

For setting up co-culturing, we first coated the cell culture chamber of a device with collagen (Corning # 354249) to provide an extracellular matrix. We seeded about 40,000 colorectal cancer cells (HCT-116) in the cell culture chamber of the device and cultured them in McCoy’s 5A medium containing 10% FBS at 37 °C and 5% CO_2_. Media was fed at 5 µL/min through side channels by a syringe pump while the inlet and outlet of the middle cell culture chamber were blocked with pegs. Once cells formed a monolayer, bacterial cells (1000 cells) or LPS (10 ng/mL) were introduced through the side channel, diffused to a cell culture chamber, and cultured in the same environmental conditions as the cancer cell culture. 

### 2.3. Quantification

A Zeiss fluorescent microscope was used to record all the images at 20× resolution. Image J (an image processing software) was used to quantify the individual cells from two–three areas for each chip, set up in duplicate. The images were converted to RGB and 8-bit format before processing and then converted to binary. The cell count and area were calculated using the particle analysis function. The data was represented as an average with a standard error. A paired Student *t*-test was used to calculate the significance.

## 3. Results

We characterized the function of the device. We first simulated the generation of the concentration gradient in the device using COMSOL Multiphysics. For simulation, we set a flow rate of 5 µL/min, a velocity of 2.1 × 10^−4^ m/s at each inlet, and a diffusivity of 5.4 × 10^−10^ m^2^/s by considering materials (e.g., oxygen scavenger and drug) with a molecular weight of about 32–400 Da. Concentrations at one side channel, cell culture chamber, and the other side channel were set at 1.0, 0.5, and 0 mol/m^3^, respectively. We obtained three distinct zones of concentration gradient with a stepwise increase from the simulation (Figure 1D). Zones along with a-a’, b-b’, and c-c’ lines represent concentration profiles at an inlet, a middle, and an outlet, respectively. The zone at the inlet formed the average concentration (Figure 1E). The zone in the middle formed three distinct gradients of the concentration with a stepwise increase (Figure 1F). The zone at the outlet formed a smooth increasing gradient of the concentration (Figure 1G). Further, we investigated the effect of flow rate on a concentration gradient profile along with the crosslines of the cell culture chamber. As the flow rate increased, we obtained a more stepwise increasing profile of concentration. We also conducted the experimental function test using food dye (Figure 1B,C). Our experimental flow pattern in the device showed a similar pattern that was obtained from the simulation. The result of the characterization of our device can be used for finding the correlation between the zonation of concentration and cell behaviors in the future. 

After we characterized the function of the device, the microfluidic device was used to test the causal effect of bacteria on cancer cell growth. In this study, we selected one example of a Gram-positive (Bacillus) and Gram-negative (LPS, a purified bacterial membrane) stimulus to mimic the conditions of the gut microbiome. A cell monolayer of cancer cells was established in the cell culture chamber. To replicate the hypoxic microenvironment of a tumor, about 1% oxygen concentration was established according to a previous procedure [39] by introducing culture media supplemented with a concentration of 1% sulfite and 100 µM cobalt for the removal of oxygen. Next, approximately 1000 Bacillus cells were loaded into the side chambers, diffused to the cell culture chamber, and co-cultured with cancer cells. Culture media with lipopolysaccharide (LPS) of 10 ng/mL was used as a positive control for inflammatory response [40]. The cell morphology and growth were quantified over time using a fluorescent microscope.

We found that both LPS and Bacillus accelerated the growth of cancer cells compared to vehicle-treated cancer cells over time. After four days of culturing, LPS promoted the growth of cancer cells 2.06-fold (*p*-value = 0.0012), while bacterial cells increased the growth 1.76-fold (*p*-value = 0.002) as compared to vehicle-treated cells (Figure 2 and Figure 3A). The same trend of significant growth increase by LPS and Bacillus bacteria was also observed when we quantified the area covered by cell growth in a microfluidic device (Figure 3B). We also noticed that the rate of cancer cell growth with bacteria alone reached a saturation point by day four, but this was not observed in LPS-treated conditions. This might be because the abundance of Bacillus decreased with time due to culture conditions, while the concentration of LPS can be maintained constant. Overall, these results demonstrated that increases in certain bacteria or microbial inflammatory stress promote colorectal cancer cell growth. Furthermore, we provided a proof-of-concept that the microfluidic chip can be used to test the causal relationship between bacteria and cancer cell growth.

## 4. Discussion

Previously several microfluidics devices have been developed to study the cancer progression [30,31] and drug testing [32,33], but only a few models are available that tested the co-culturing of bacteria and cancer cells in hypoxic conditions. Here we developed a simplistic co-culturing microfluidics model that allows the maintenance of hypoxic conditions and maintains media flow to mimic gut physiological conditions. Using the device, we demonstrated that the abundance of bacteria could significantly promote the growth of cancer cells. However, colorectal cancer is highly abundant in various Gram-positive (Streptococcus, Gemella) microflora with few Gram-negative (Fusobacterium) genus representations [38]. For each of these genera, multiple bacterial species could be associated. Therefore, the testing of multiple bacterial types was beyond the scope of this study but should be tested in the future. There are several possible mechanisms by which an imbalance in the bacterial community can impact cancer cell growth. Most bacteria secrete substances, such as LPS and peptidoglycan, which are found in the bacterial cell membrane [8,40]. When these substances enter the gut and bind to the receptors on the myeloid cells found therein, these interactions cause epigenetic reprogramming leading to changes in the gene expression profile. This can lead to increased pro-inflammatory cytokine secretion, reduced clearance of pathogens and cancer cells, and reduced intestinal barrier fortification, allowing more pathogens to enter [3].

LPS and peptidoglycan also increase the secretion of pro-inflammatory cytokines, increasing cancer progression. When LPS and peptidoglycan bind the toll-like receptor (TLR2, TLR4) on the cell membrane, the receptor signals to the IRAK kinase, which activates the NF-kB1 transcription factor leading to the increased secretion of pro-inflammatory cytokines, such as IL6 [7,8]. This TLR pathway can be targeted by drugs, such as IRAK-1 kinase and NF-kB1 inhibitors or IL-6 blocking antibodies, to reduce cancer progression [7,8,41]. Recent studies suggest that an improved understanding of the mechanism and causal relationship between bacteria and cancer cells can be harnessed to develop new therapeutic strategies against cancer [11]. The microfluidic device utilized in this study has broader implications, including simulating other cancer types. This device can test both the effect of bacteria and new treatments on clinical samples to identify personalized therapy, thus reducing the need for a mouse model for preclinical testing, which is a lengthy and expensive process. These devices are also disposable and cost-effective. In summary, we established a new co-culture system for bacteria and cancer cells and identified that the human gut microbiome could promote cancer cell growth. Our study suggests that the microfluidic device can also be used to test a new therapeutic entity in cancer research.

## Figures and Tables

**Figure 1 sensors-23-01265-f001:**
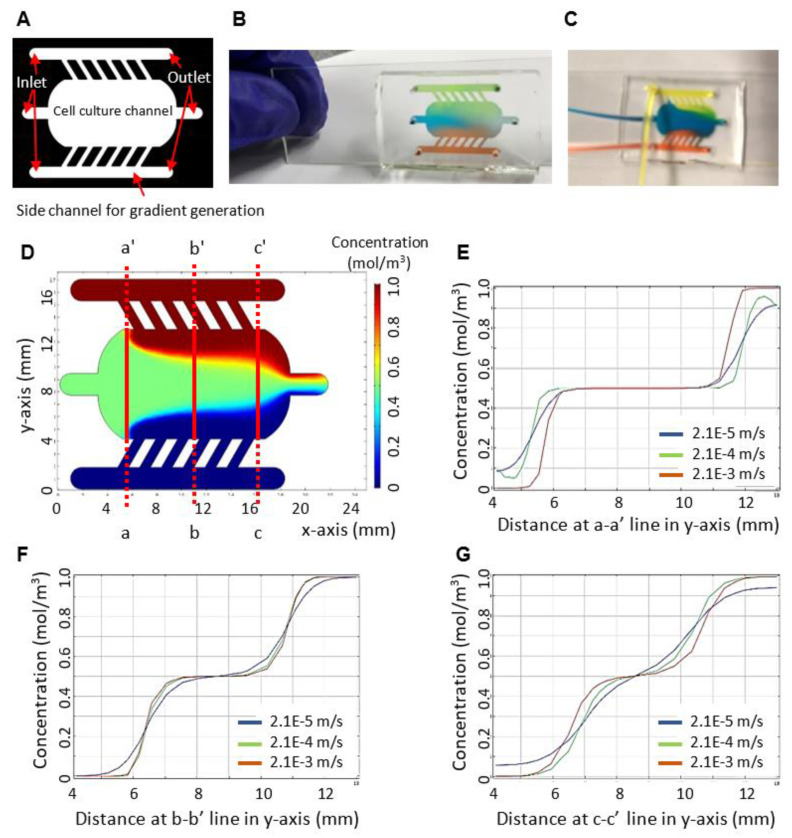
The characterization of the microfluidic device. (**A**) The device consists of a cell culture chamber and side channels for gradient generation. The design of the device allows to co-culture of bacteria and cancer cells. (**B**,**C**) The functional tests of the device using food dye at static and dynamic conditions. (**D**) The characterization of a concentration gradient in the gut chip through simulation by COMSOL Multiphysics. A velocity of an inlet is 2.1 × 10^−4^ m/s. Diffusion coefficient is 5.4 × 10^−10^ m^2^/s. Concentrations at one side channel, cell culture chamber, and the other side channel are 1.0, 0.5, and 0 mol/m^3^, respectively. (**E**) The effect of flow rate on a concentration gradient profile along with the crossline a-a’ of the cell culture chamber. (**F**) The effect of flow rate on a concentration gradient profile along with the crossline b-b’ of the cell culture chamber. (**G**) The effect of flow rate on a concentration gradient profile along with the crossline c-c’ of the cell culture chamber.

**Figure 2 sensors-23-01265-f002:**
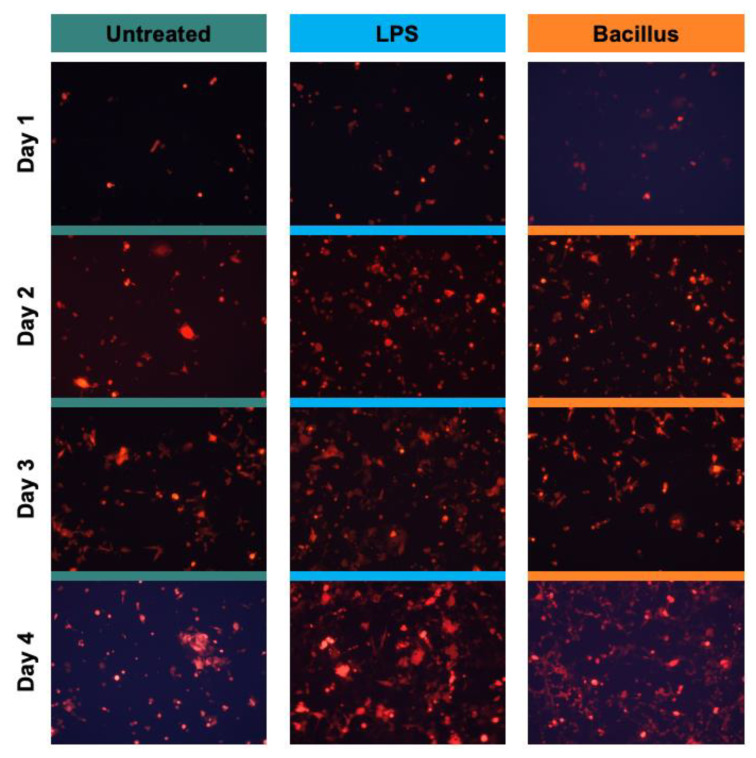
The bacterial stimulus promotes the growth of colorectal cancer cells over time. In the outer inlets of the device, approximately 1000 bacterial cells were loaded. About 40,000 HCT 116 cells labeled with a red fluorescent protein (RFP) were injected into the middle inlet. Cells were cultured for 24 hrs with Bacillus or lipopolysaccharide (LPS; 10 ng/mL). Then microchannels were washed thoroughly to remove non-adherent HCT 116 cells and extra bacterial cells. Media was fed at a 5 µL/min flow rate. These experiments were performed without hypoxic conditions. A representative image for each time point is shown.

**Figure 3 sensors-23-01265-f003:**
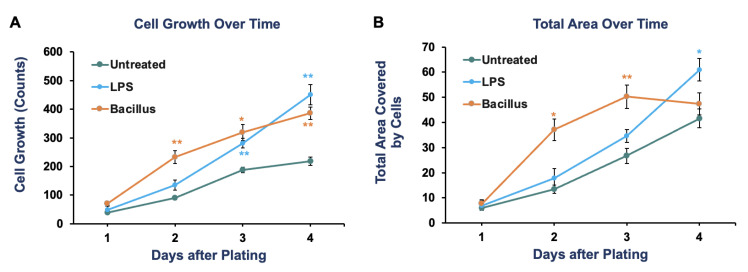
The effect of Bacillus or lipopolysaccharide (LPS; 10 ng/mL) on the growth of colorectal cancer cells (HCT 116) over time is quantified daily by imaging two to three independent areas of a cell culture chamber. The number of cells in images was counted using Image J. The data represents the average quantification with a standard error of 4–5 images as cell counts (**A**) and area covered by cells (**B**). These experiments were performed without hypoxic conditions. Student *t*-test was used to calculate the *p*-value. * *p* < 0.05 and ** *p* < 0.01.

## Data Availability

All data generated or analyzed during this study are included in this published article or can be obtained by contacting the authors.

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
