# Peer review of "Simulating the Effect of Gut Microbiome on Cancer Cell Growth Using a Microfluidic Device"

_sensors, 2023, doi:10.3390/s23031265_

Round 1
Reviewer 1 Report
The authors developed a microfluidic chip that simulate the interaction of the gut microbiome and cancer cells to investigate the effects of bacteria and inflammatory stress on cancer cells in vitro. However, the manuscript is lack of novelty and the conclusions need to be supported by additional experiments.
Item 1. The introduction did not provide sufficient background about the microfluidic devices such as organ-on-a-chip.
Item 2. The authors did not provide the advantage of using microfluidic device. There is no specific structure for the microfluidic chip. The only use for the chip in the manuscript for me is to provide cell culture space, please show the difference between the culture dish and the chip.
Item 3. Why did the authors choose Bacillus? “Bacillus was selected due to BSL-1 safety level” is not a scientific reason. More bacteria need to be considered.
Item 4. Did the concentration differences of Bacillus effect the result? The design of the experiment is too simple without negative and positive controls.
Item5. To solidate the conclusion, more duplicated experiments need to be done.
Item 5. The resolution of Fig.2 and Fig.3 has to be improved.
Overall, the paper should no be accepted by the journal at this current format.
Author Response
"Please see the attachment."

Reviewer 2 Report
The article is devoted to an important aspect of research, difficult to carry out in vivo, but very useful and desirable. The development of this type of microfluidic devices may enable the determination of the influence of various factors on the development of pathogenic microorganisms. For this reason, this direction of research is very desirable. The authors could specify whether the chip is disposable or not.
Reviewer 3 Report
In this study, the authors claim the fabrication of a new co-culture system for bacteria and cancer cells and outlined that the human gut microbiome can promote the growth of cancer cells. I strongly believe that the article is not suitable for publication in its current form due to the concerns listed below.
1) First of all, this study is not dealing with any sensing or detection principle - hence, it hardly falls into the scope of the journal!
2) There are a lot of studies reporting the co-culture systems, see for example these works: doi:10.1038/srep35544 and doi: 10.3791/54344
3) An introduction part and the manuscript are too short - should be expanded by recent studies in this area. In particular, highlight what has been done before and what is the novelty of this study.
4) Results are too simplistic and do not take into account the complex nature of the gut microbiome.
5) It is suggested to investigate the real mechanism(s) that impact the growth of cancer cells rather than listing them.
Round 2
Reviewer 1 Report
Accept in present form.
Reviewer 3 Report
A revised manuscript can be considered for publication.